# In-Place Zero-Space Memory Protection for CNN

**Hui Guan**[1], **Lin Ning**[1], **Zhen Lin**[1], **Xipeng Shen**[1], **Huiyang Zhou**[1], **Seung-Hwan Lim**[2]

[1]North Carolina State University, Raleigh, NC, 27695
[2]Oak Ridge National Laboratory, Oak Ridge, TN 37831
{hguan2, lning, zlin4, xshen5, hzhou}@ncsu.edu, lims1@ornl.gov

## Abstract

Convolutional Neural Networks (CNN) are being actively explored for safety-critical applications such as autonomous vehicles and aerospace, where it is essential to ensure the reliability of inference results in the presence of possible memory faults. Traditional methods such as error correction codes (ECC) and Triple Modular Redundancy (TMR) are CNN-oblivious and incur substantial memory overhead and energy cost. This paper introduces *in-place zero-space ECC* assisted with a new training scheme *weight distribution-oriented training*. The new method provides the first known zero space cost memory protection for CNNs without compromising the reliability offered by traditional ECC.

## 1 Introduction

As CNNs are increasingly explored for safety-critical applications such as autonomous vehicles and aerospace, reliability of CNN inference is becoming an important concern. A key threat is memory faults (e.g., bit flips in memory), which may result from environment perturbations, temperature variations, voltage scaling, manufacturing defects, wear-out, and radiation-induced soft errors. These faults change the stored data (e.g., CNN parameters), which may cause large deviations of the inference results [13, 19, 20]. In this work, fault rate is defined as the ratio between the number of bit flips experienced before correction is applied and the total number of bits.

Existing solutions have resorted to general memory fault protection mechanisms, such as Error Correction Codes (ECC) hardware [24], spatial redundancy, and radiation hardening [28]. Being CNN-oblivious, these protections incur large costs. ECC, for instance, uses eight extra bits in protecting 64-bit memory; spatial redundancy requires at least two copies of CNN parameters to correct one error (called Triple Modular Redundancy (TMR) [14]); radiation hardening is subject to substantial area overhead and hardware cost. The spatial, energy, and hardware costs are especially concerning for safety-critical CNN inferences; as they often execute on resource-constrained (mobile) devices, the costs worsen the limit on model size and capacity, and increase the cost of the overall AI solution.

To address the fundamental tension between the needs for reliability and the needs for space/energy/cost efficiency, this work proposes the first zero space cost memory protection for CNNs. The design capitalizes on the opportunities brought by the distinctive properties of CNNs. It further amplifies the opportunities by introducing a novel training scheme, *Weight Distribution-Oriented Training (WOT)*, to regularize the weight distributions of CNNs such that they become more amenable for zero-space protection. It then introduces a novel protection method, *in-place zero-space ECC*, which removes all space cost of ECC protection while preserving protection guarantees.

Experiments on VGG16, ResNet18, and SqueezeNet validate the effectiveness of the proposed solution. Across all tested scenarios, the method provides protections consistently comparable to those offered by existing hardware ECC logic, while removing all space costs. It hence offers a promising replacement of existing protection schemes for CNNs.

## 2   Related Work

There are some early studies on fault tolerance of earlier neural networks (NN) [16, 17, 26]; they examined the performance degradation of NNs with various fault models on networks that differ from modern CNNs in both network topologies and and model complexities.

Fault tolerance of deep neural networks (DNN) has recently drawn increasing attentions. Li et al. [13] studied the soft error propagation in DNN accelerators and proposed to leverage symptom-based error detectors for detecting errors and a hardware-based technique, *selective latch hardening*, for detecting and correcting data-path faults. A recent work [3, 19] conducted some empirical studies to quantify the fault tolerance of DNNs to memory faults and revealed that DNN fault tolerance varies with respect to model, layer type, and structure. Zhang et al. [30] proposed fault-aware pruning with retraining to mitigate the impact of permanent faults for systolic array-based CNN accelerators (e.g., TPUs). They focused only on faults in the data-path and ignored faults in the memory. Qin et al. [18] studied the performance degradation of 16-bit quantized CNNs under different bit flip rates and proposed to set values of detected erroneous weights as zeros to mitigate the impact of faults. These prior works focused mainly on the characterization of DNN's fault tolerance with respect to various data types and network topologies. While several software-based protection solutions were explored, they are preliminary. Some can only detect but not correct errors (e.g. detecting extreme values [13]), others have limited protection capability (e.g. setting faulty weights to zero [18]).

Some prior work proposes designs of energy-efficient DNN accelerators by exploiting fault tolerance of DNNs [12, 25, 29]. An accelerator design [20] optimizes SRAM power by reducing the supply voltage. It leverages active hardware fault detection coupled with bit masking that shifts data towards zero to mitigate the impact of bit flips on DNNs' model accuracy without the need of re-training. Similar hardware faults detection techniques are later exploited in [7, 22, 27, 29] to improve fault tolerance of DNNs. Azizimazreah et al. [4] proposed a novel memory cell designed to eliminate soft errors while achieving a low power consumption. These designs are for some special accelerators rather than general DNN reliability protection. They are still subject to various costs and offer no protection guarantees as existing ECC protections do. This current work aims to reducing the space cost of protection to zero without compromising the reliability of existing protections.

## 3   Premises and Scopes

This work focuses on protections of 8-bit quantized CNN models. On the one hand, although the optimal bit width for a network depends on its weight distribution and might be lower than 8, we have observed that 8-bit quantization is a prevalent, robust, and general choice to reduce model size and latency while preserving accuracy. In our experiments, both activations and weights are quantized to 8-bit. Existing libraries that support quantized CNNs (e.g. NVIDIA TensorRT [15], Intel MKL-DNN [1], Google's GEMMLOWP [10], Facebook's QNNPACK [2]) mainly target for fast operators using 8-bit instead of lower bit width. On the other hand, previous studies [13, 19] have suggested that CNNs should use data types that provide just-enough numeric value range and precision to increase its fault tolerance. Our explorations on using higher precision including float32 for representing CNN parameters also show that 8-bit quantized models are the most resilient to memory faults.

The quantization algorithm we used is symmetric range-based linear quantization that is well-supported by major CNN frameworks (e.g. Tensorflow [11], Pytorch [32]). Specifically, let $X$ be a floating-point tensor and $X^q$ be the 8-bit quantized version. $X$ can be either weights or activations from a CNN. The quantization is based on the following formula:

$$X^q = round(X \frac{2^{n-1} - 1}{\max\{|X|\}}),  \tag{1}$$

where $n$ is the number of bits used for quantization. In our case, $n = 8$. The number of bits used for accumulation is 32. Biases, if exist, are quantized to 32 bit integer.

Our work protects only weights for two reasons. Firstly, weights are usually kept in the memory. The longer they are kept, the higher the number of bit flips they will suffer from. This easily results in a high fault rate (e.g. 1e-3) for weights. Activations, however, are useful only during an inference process. Given the slight chance of having a bit flip during an inference process (usually

Table 1: Accuracy and weight distribution of 8-bit quantized CNN models on ImageNet. The percentage rows use absolute values.

| Model | | AlexNet | VGG16 | VGG16_bn | Inception_V3 | ResNet18 | ResNet34 | ResNet50 | ResNet152 | SqueezeNet |
|---|---|---|---|---|---|---|---|---|---|---|
| #weights | | 61.1M | 138.4M | 138.4M | 27.1M | 11.7M | 21.8M | 25.5M | 60.1M | 1.2M |
| Accuracy (%) | Float32 | 56.52 | 71.59 | 73.36 | 69.54 | 69.76 | 73.31 | 76.13 | 78.31 | 58.09 |
| | Int8 | 55.8 | 71.51 | 72.01 | 68.07 | 69.07 | 72.83 | 75.33 | 77.79 | 57.01 |
| Percentage (%) | [0, 32) | 95.09 | 97.69 | 98.83 | 97.98 | 99.66 | 99.76 | 99.65 | 99.49 | 95.16 |
| | [32, 64) | 4.88 | 2.27 | 1.16 | 1.96 | 0.32 | 0.23 | 0.34 | 0.49 | 4.62 |
| | [64, 128] | 0.03 | 0.04 | 0.01 | 0.06 | 0.02 | 0.01 | 0.01 | 0.01 | 0.22 |

in milliseconds), protecting activations is not as pressing as protecting weights. Secondly, previous work [19] has shown that activations are much less sensitive to faults compared with weights.

Error Correction Codes (ECC) is commonly used in computer systems to correct memory faults. They are usually described as $(k, d, t)$ code for length $k$ code word, length $d$ data, and $t$-bit error correction. The number of required check bits is $k - d$.

## 4 In-Place Zero-Space ECC

Our proposed method, in-place zero-space ECC, builds on the following observation:*Weights of a well-trained CNN are mostly small values.* The *Percentage* rows in Table 1 show the distributions of the absolute values of weights in some popular 8-bit quantized CNN models. The absolute values of more than 99% of the weights are less than 64. Even though eight bits are used to represent each weight, if we already know that the absolute value of a weight is less than 64, the number of effective bits to represent the value would be at most seven, and the remaining bit could be possibly used for other purposes—such as error correction. We call it a *non-informative bit*.

The core idea of in-place zero-space ECC is to use *non-informative bits* in CNN parameters to store error check bits. For example, the commonly used SEC-DED $(64, 57, 1)$ code uses seven check bits to protect 57 data bits for single error correction; they together form a 64-bit code word. If seven out of eight consecutive weights are in range $[-64, 63]$, we can then have seven non-informative bits, one per small weight. The essential idea of in-place ECC is to use these non-informative bits to store the error check bits for the eight weights. By embedding the check bits into the data, it can hence avoid all space cost.

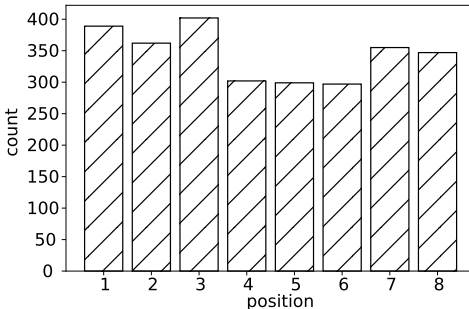

Figure 1: Large weight (beyond $[-64, 63]$) distributions in 8-byte (64-bit data) blocks for SqueezeNet on ImageNet. For instance, the first bar in (a) shows that of all the 8-byte data blocks storing weights, around 380 have a large weight at the first byte.

For the in-place ECC to work, there cannot be more than one large weight in every 8 consecutive weights. And the implementation has to record the locations of the large weights such that the decoding step can find the error check bits from the data. It is, however, important to note that the requirement of recording the locations of large weights would disappear if the large weights are regularly distributed in data—an example is that the only place in which a large weight could appear is the last byte of an 8-byte block. However, the distributions of large weights in CNNs are close to uniform, as Figure 1 shows.

### 4.1 WOT

To eliminate the need of storing large weight locations in in-place ECC, we enhance our design by introducing a new training scheme, namely *weight-distribution oriented training (WOT)*. WOT aims to regularize the spatial distribution of large weights such that large values can appear only at specific places. We first formalize the WOT problem and then elaborate our regularized training process.

Let $W_l$ be the float32 parameters (including both weights and biases) in the $l$-th convolutional layer and $W_l^q$ be their values after quantization. Note that WOT applies to fully-connected layers as well even though our discussion focuses on convolutional layers. WOT minimizes the sum of the standard cross entropy loss ($f(\{W_l^q\}_{l=1}^L)$) and weighted weight regularization loss (Frobenius norm with the hyperparameter $\lambda$) subject to some weight distribution constraints on the weights:

$$\min_{\{W_l^q\}} f(\{W_l^q\}_{l=1}^L) + \lambda \sum_{l=1}^L \|W_l^q\|_F^2, \tag{2}$$

$$s.t. \quad W_l^q \in S_l, l = 1, \cdots, L. \tag{3}$$

The weights are a four-dimensional tensor. If flattened, it is a vector of length $N_l \times C_l \times H_l \times W_l$, where $N_l$, $C_l$, $H_l$ and $W_l$ are respectively the number of filters, the number of channels in a filter, the height of the filter, and the width of the filter, in the $l$-th convolutional layer. WOT adds constraints to each 64-bit data block in the flattened weight vectors. Recall that, for in-place ECC to protect a 64-bit data block, we need seven non-informative bits (i.e., seven small weights in the range $[-64, 63]$) to store the seven check bits. To regularize the positions of large values in weights, the constraint on the weights in the $l$-th convolutional layer can be given by $S_l = \{X|$ the first seven values in every 64-bit data block can have a value in only the range of $[-64, 63]\}$.

We next describe two potential solutions to the optimization problems.

**ADMM-based Training**  The above optimization problem can be formulated in the Alternating Direction Method of Multipliers (ADMM) framework and solved in a way similar to an earlier work [31]. The optimization problem (Eq. 2) is equivalent to:

$$\min_{\{W_l^q\}} f(\{W_l^q\}_{l=1}^L) + \lambda \sum_{l=1}^L \|W_l^q\|_F^2 + \sum_{l=1}^L g_l(W_l^q), \tag{4}$$

where $g_l(W_l^q) = \begin{cases} 0, & \text{if } W_l^q \in S_l \\ +\infty, & \text{otherwise.} \end{cases}$. Rewriting Eq. 4 in the ADMM framework leads to:

$$\min_{\{W_l^q\}} f(\{W_l^q\}_{l=1}^L) + \lambda \sum_{l=1}^L \|W_l^q\|_F^2 + \sum_{l=1}^L g_l(Z_l), \tag{5}$$

$$s.t. \quad W_l^q = Z_l, l = 1, \cdots, L \tag{6}$$

ADMM alternates between the optimization of model parameters ($\{W_l^q\}_{l=1}^L$ and the auxiliary variables $\{Z_l\}_{l=1}^L$ by repeating the following three steps for $k = 1, 2, \cdots$ :

$$\{W_l^{q,k+1}\}_{l=1}^L = \arg \min_{\{W_l^q\}_{l=1}^L} f(\{W_l^q\}_{l=1}^L) + \lambda \sum_{l=1}^L \|W_l^q\|_F^2 + \gamma \sum_{l=1}^L \|W_l^q - Z_l + U_l^k\|_F^2, \tag{7}$$

$$\{Z_l^{k+1}\}_{l=1}^L = \arg \min_{\{Z_l\}_{l=1}^L} \sum_{l=1}^N g_l(Z_l) + \sum_{l=1}^L \frac{\lambda}{2} \|W_l^{q,k+1} - Z_l + U_l^k\|_F^2, \tag{8}$$

$$U_l^{k+1} = U_l^k + W_l^{q,k+1} - Z_l^{k+1}. \tag{9}$$

until the two conditions are met: $\|W_l^{q,k+1} - Z_l^{k+1}\|_F^2 \leq \epsilon$ and $\|Z_l^{k+1} - Z_l^k\|_F^2 \leq \epsilon$.

Problem 7 can be solved using stochastic gradient descent (SGD) as the objective function is differentiable. The optimal solution to the problem 8 is the projection of $W_l^{q,k+1} + U_l^k$ to set $S_l$. In the implementation, we set a value in a 64-data block to 63 or -64 if the value is not in the eighth position and is larger than 63 or smaller than -64.

Previous work has successfully applied the ADMM framework to CNN weight pruning [31] and CNN weight quantization [21] and shown remarkable compression results. But when it is applied to our problem, experiments show that ADMM-based training cannot help reduce the number of large values in the first seven positions of a 64-bit data block. Moreover, as the ADMM-based training cannot guarantee that the constrain in Eq. 3 is satisfied, it is necessary to bound the reamining large quantized values in the first 7 positions to 63 or -64 after the training, resulting in large accuracy drops. Instead of ADMM-based training, WOT adopts an alternative approach described below.

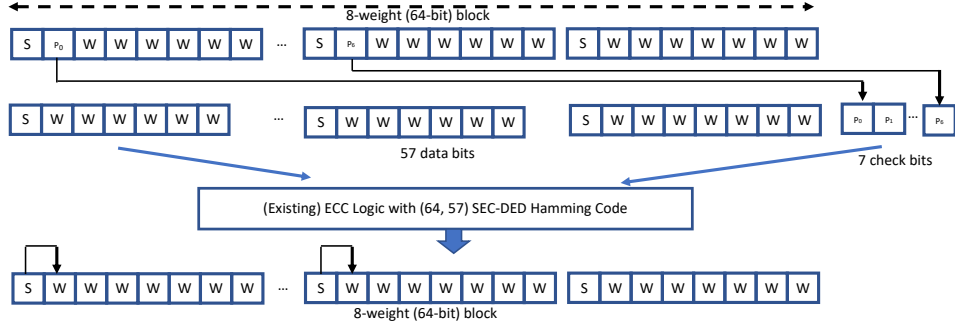

Figure 2: Hardware design for in-place zero-space ECC protection.

**QAT with Throttling (QATT)**    Our empirical explorations indicate that a simple quantization-aware training (QAT) procedure combined with weight throttling can make the weights meet the constraint without jeopardising the accuracy of a 8-bit quantized model. The training process iterates the following major steps for each batch:

1. **QAT:** It involves forward-propagation using quantized parameters ($\{W_l^q\}_{l=1}^L$ and $\{b_l^q\}_{l=1}^L$) to get the loss defined in Equation 2, back-propagation using quantized parameters, a update step that applies float32 gradients to update float32 parameters ($\{W_l\}_{l=1}^L$ and $\{b_l\}_{l=1}^L$), and a quantization step that gets the new quantized parameters from their float32 version.

2. **Throttling:** It forces the quantized weights to meet the constraints defined in Eq. 3: If any value in the first seven bytes of a 64-bit data block is larger than 63 (or less than -64), set the value to 63 (or -64). The float32 versions are updated accordingly.

After the training, all of the values in the first seven positions of a 64-bit data block are ensured to be within the range of $[-64, 63]$, eliminating the need of storing large value positions for the in-place ECC. It is worth noting that with WOT, all tested CNNs converge without noticeable accuracy loss compared to the 8-bit quantized versions as Section 5 shows.

### 4.2   Full Design of In-Place Zero-Space ECC

In this part, we provide the full design of *in-place zero-space ECC*. For a given CNN, it first applies WOT to regularize the CNN. After that, it conducts in-place error check encoding. The encoding uses the same encoding algorithm as the standard error-correction encoding methods do; the difference lies only in where the error check bits are placed.

There are various error-correction encoding algorithms. In principle, our proposed in-place ECC could be generalized to various codes; we focus our implementation on SEC-DED codes for its popularity in existing hardware-based memory protections for CNN.

Our in-place ECC features the same protection guarantees as the popular SEC-DED $(72, 64, 1)$ code but at zero-space cost. The in-place ECC uses the SEC-DED $(64, 57, 1)$ code instead of $(72, 64, 1)$ to protect a 64-bit data block with the same protection strength. It distributes the seven error check bits into the non-informative bits in the first seven weights.

As the ECC check bits are stored in-place, a minor extension to the existing ECC hardware is required to support ECC decoding. As shown in Figure 2, the in-place ECC check bits and data bits are swizzled to the right inputs to the standard ECC logic. The output of the ECC logic is then used to recover the original weights: for each small weight (first seven bytes in a 8-byte data block), simply copy the sign bit to its non-informative bit. As only additional wiring is needed to implement this copy operation, no latency overhead is incurred to the standard ECC logic.

## 5   Evaluations

We conducted a set of experiments to examine the efficacy of the proposed techniques in fault protection and overhead. We first describe our experiment settings in Section 5.1 and then report the effects of WOT and the proposed fault protection technique in Sections 5.2 and 5.3.

## 5.1 Experiment Settings

**Models, Datasets, and Machines** The models we used in the fault injection experiments include VGG16 [23], ResNet18 [8], and SqueezeNet [9]. We choose these CNN models as representatives because: 1) VGG is a typical CNN with stacked convolutional layers and widely used in transfer learning because of its robustness. 2) ResNets are representatives of CNNs with modular structure (e.g. Residual Module) and are widely used in advanced computer vision tasks such as object detection. 3) SqueezeNet has much fewer parameters and represents CNNs that are designed for mobile applications. The accuracies of these models are listed in Table 1. By default, We use the ImageNet dataset [6] (ILSVRC 2012) for model training and evaluation. All the experiments are performed with PyTorch 1.0.1 on machines equipped with a 40-core 2.2GHz Intel Xeon Silver 4114 processor, 128GB of RAM, and an NVIDIA TITAN Xp GPU with 12GB memory. Distiller [32] is used for 8-bit quantization. The CUDA version is 10.1.

**Counterparts for Comparisons** We compare our method (denoted as **in-place**) with the following three counterparts:

- **No Protection (faulty):** The CNN has no memory protection.
- **Parity Zero (zero):** It adds one parity bit to detect single bit errors in an eight-bit data block (e.g. a single weight parameter). Once errors are detected, the weight is set to zero[1].
- **SEC-DED (ecc)** It is the traditional SEC-DED [72, 64, 1] code-based protection in computer systems [24].

There are some previous proposals [4, 20] of memory protections, which are however designed for special CNN accelerators and provide without protection guarantees. The parity and ECC represent the state of the art in the industry for memory protection that work generally across processors and offer protection guarantees, hence the counterparts for our comparison.

## 5.2 WOT results

We evaluate the efficiency of WOT using the CNNs shown in Table 1. All the models are pre-trained on ImageNet (downloaded from TorchVision[2]). We set $\lambda$ to 0.0001 for all of the CNNs. Model training uses stochastic gradient descent with a constant learning rate 0.0001 and momentum 0.9. Batch size is 32 for VGG16_bn and ResNet152, 64 for ResNet50 and VGG16, and 128 for the remaining models. Training stops as long as the model accuracy after weight throttling reaches its 8-bit quantized version.

Figure 3 shows the changes of the total number of large values that are beyond $[-64, 63]$ in the first seven positions of 8-byte blocks during the training on six of the CNNs. WOT successfully reduces this number from more than 3,500–80,000 to near 0 for the models before throttling during the training process. The remaining few large values in non-eighth positions are set to -64 or 63 at the end of WOT. Note that VGG16_bn has around 10000 large values in the non eighth positions after 8k iterations. Although more iterations further reduce this number, VGG16_bn can already reach its original accuracy after weight throttling.

The accuracy curves of the models in the WOT training are shown in Figure 4. Overall, after WOT training, the original accuracy of all the six networks are fully recovered. During the training, the gap between the accuracy before throttling and after throttling is gradually reduced. For example, the top-1 accuracy of SqueezeNet after 8-bit quantization is 57.01%. After the first iteration of WOT, the accuracy before weight throttling is 31.38% and drops to 11.54% after throttling. WOT increases the accuracy to 57.11% after 46k iterations with batch size 128 (around 4 epochs). All the other CNNs are able to recover their original accuracy in only a few thousands of iterations. An exception is VGG16, which reaches an accuracy of 71.50% (only 0.01% accuracy loss) after 20 epochs of training.

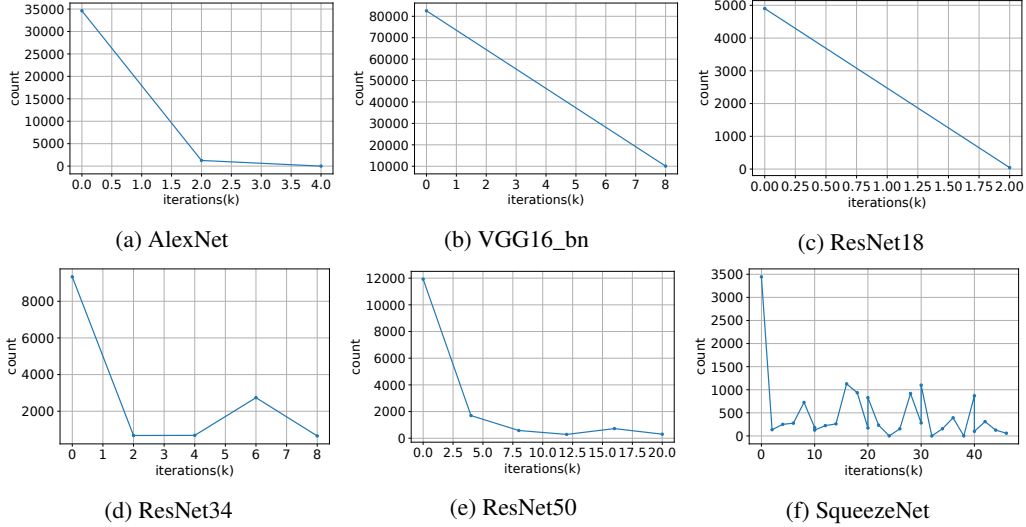

Figure 3: Changes of the total number of large values (beyond $[-64, 63]$) in the first 7 positions of 8-byte (64-bit data) blocks before the throttling step during the WOT training process.

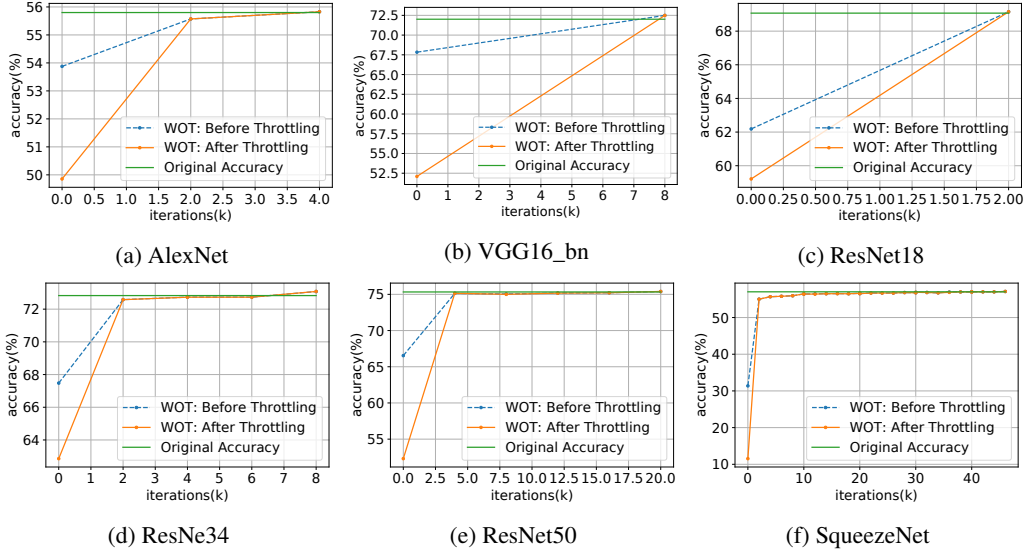

Figure 4: Accuracy curves before and after the throttling step during the WOT training process.

## 5.3 Fault injection results

In this set of experiments, we inject faults to CNN models and report the accuracy drops of CNN models protected using different strategies. The fault model is random bit flip. Faults are injected to the weights of CNNs with memory fault rates varying from $10^{-9}$ to $0.001$. The number of faulty bits is the product of the number of bits used to represent weights of a CNN and the memory fault rate. We repeated each fault injection experiment ten times.

Table 2 shows the mean accuracy drops with standard deviations under different memory fault rates and the overheads introduced by the protection strategies for each model. Overall, the in-place ECC protection and standard SEC-DED show similar accuracy drop patterns under various fault rate settings as expected because they provide the same error correction capability, i.e., correcting a single bit error and detecting double bit errors in a 64-bit data block. Both of the methods provide stronger fault protection compared with the Parity Zero method. The space overhead is the ratio between the extra number of bytes introduced by a protection strategy and the number of bytes required to store weights. Parity Zero and SEC-DED encode 8-byte data with extra eight check bits on average, making their space overhead 12.5%. In contrast, in-place ECC has zero space cost.

Table 2: Accuracy drop of VGG16, ResNet16, and SqueezeNet under different memory fault rates.

| Model | Strategy | ECC HW (Y/N) | Space Overhead (%) | Accuracy drop (%) under different fault rate | | | |
|---|---|---|---|---|---|---|---|
| | | | | 1e-06 | 1e-05 | 1e-04 | 1e-03 |
| VGG16 | faulty | N | 0 | $0.31 \pm 0.08$ | $0.47 \pm 0.09$ | $1.35 \pm 0.2$ | $21.93 \pm 5.7$ |
| | zero | N | 12.5 | $0.27 \pm 0.05$ | $0.36 \pm 0.08$ | $0.43 \pm 0.13$ | $1.04 \pm 0.31$ |
| | ecc | Y | 12.5 | $0.0 \pm 0.0$ | $0.02 \pm 0.02$ | $0.35 \pm 0.06$ | $0.96 \pm 0.14$ |
| | in-place | Y | 0 | $0.0 \pm 0.0$ | $0.02 \pm 0.02$ | $0.37 \pm 0.07$ | $0.93 \pm 0.23$ |
| ResNet18 | faulty | N | 0 | $-0.09 \pm 0.1$ | $0.35 \pm 0.23$ | $4.35 \pm 1.12$ | $72.96 \pm 1.48$ |
| | zero | N | 12.5 | $-0.06 \pm 0.08$ | $-0.08 \pm 0.13$ | $0.59 \pm 0.3$ | $4.35 \pm 1.21$ |
| | ecc | Y | 12.5 | $0.0 \pm 0.0$ | $0.0 \pm 0.01$ | $-0.03 \pm 0.08$ | $2.8 \pm 0.31$ |
| | in-place | Y | 0 | $0.0 \pm 0.0$ | $0.0 \pm 0.01$ | $-0.08 \pm 0.09$ | $2.96 \pm 0.81$ |
| SqueezeNet | faulty | N | 0 | $0.12 \pm 0.13$ | $0.69 \pm 0.31$ | $9.39 \pm 2.37$ | $64.83 \pm 0.5$ |
| | zero | N | 12.5 | $0.09 \pm 0.12$ | $0.11 \pm 0.2$ | $0.66 \pm 0.29$ | $8.16 \pm 2.4$ |
| | ecc | Y | 12.5 | $0.0 \pm 0.0$ | $0.0 \pm 0.0$ | $0.12 \pm 0.09$ | $5.37 \pm 0.66$ |
| | in-place | Y | 0 | $0.0 \pm 0.0$ | $0.0 \pm 0.0$ | $0.12 \pm 0.09$ | $5.19 \pm 1.08$ |

The fault injection experiments give the following insights on memory fault protection for CNNs. First, larger models tend to suffer less from memory faults. For example, when fault rate is 0.0001 and no protection is applied, the accuracy drops of VGG16, ResNet18, SqueezeNet (less than 2%, 8%, and 16% respectively) are increasing while the model size is decreasing (number of parameters are 138M, 12M, and 1.2M respectively). Second, when the fault rate is small (e.g. less than 1e-05), in-place ECC and standard SEC-DED can almost guarantee the same accuracy as the fault-free model. Overall, the experiments confirm the potential of in-place zero-space ECC as an efficient replacement of the standard ECC without compromising the protection quality.

# 6  Future Directions

Besides 8-bit quantizations, there are proposals of even fewer-bit quantizations for CNN, in which, there may be fewer non-informative bits in weight values. It is however worth noting that 8-bit quantization is the de facto in most existing CNN frameworks; it has repeatedly shown in practice as a robust choice that offers an excellent balance in model size and accuracy. Improving the reliability of such models is hence essential. With that said, creating zero-space protections that works well with other model quantizations is a direction worth future explorations.

A second direction worth exploring is to extend the in-place zero-space protection to other error encoding methods (e.g., BCH [5]). Some of them require more parity bits, for which, the regularized training may need to be extended to create more free bits in data.

Finally, in-place zero-space ECC is in principle applicable to neural networks beyond CNN. Empirically assessing the efficacy is left to future studies.

# 7  Conclusions

This paper presents *in-place zero-space ECC* assisted with a new training scheme named WOT to protect CNN memory. The protection scheme removes all space cost of ECC without compromising the reliability offered by ECC, opening new opportunities for enhancing the accuracy, energy efficiency, reliability, and cost effectiveness of CNN-driven AI solutions.

**Acknowledgement**  We would like to thank the anonymous reviews for their helpful feedbacks. This material is based upon work supported by the National Science Foundation (NSF) under Grant No. CCF-1525609, CCF-1703487, CCF-1717550, and CCF-1908406. Any opinions, comments, findings, and conclusions or recommendations expressed in this material are those of the authors and do not necessarily reflect the views of NSF. This manuscript has been authored by UT-Battelle, LLC, under contract DE-AC05-00OR22725 with the US Department of Energy (DOE). The US government retains and the publisher, by accepting the article for publication, acknowledges that the US government retains a nonexclusive, paid-up, irrevocable, worldwide license to publish or reproduce the published form of this manuscript, or allow others to do so, for US government purposes. DOE will provide public access to these results of federally sponsored research in accordance with the DOE Public Access Plan (http://energy.gov/downloads/doe-public-access-plan).

## Footnotes

[1]We have tried to set a detected faulty weight to the average of its neighbors but found it has worse performance than Parity Zero.

[2]https://pytorch.org/docs/master/torchvision/

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
