[Reviews · NeurIPS 2019]

Reviewer 1



I really liked this paper. The idea of creating space for ECC bits within the network is both interesting and practical. The evaluation is thorough and on the whole, the paper is well written and easy to follow.

Reviewer 2



a) The paper assumes a system, which uses a specific approach for protecting DNN weights against memory faults. However, it does not mention how the system in question handles potential memory faults against its input/output tensors. Not storing input/output tensors in memory or using different strategies for memory protection of different data (weights vs. input/output) imposes conditions on the system, thereby making the solution not generally applicable. b) Section 3 of the paper asserts the following: 67 This work focuses on protections of 8-bit quantized DNN models. The reason are two-fold: 1) 68 8-bit quantization has been the de facto step before model deployment to reduce model size while 69 providing lower latency with little degradation in model accuracy. 2) Previous studies [10, 15] However, the paper does not provide any references or comparisons with lower precisions (e.g. 4-bit) to justify reason 1). The optimal quantization precision (balance accuracy and model size) for any network depends on its weight distribution, which might be lower than 8-bit as well. Lastly, making the two protection methods heavily dependent on reason 1) imposes more restrictions on the system where such techniques can be used. c) The paper demonstrates the impact of its protection techniques only for CNNs (VGG16, ResNet18, and SqueezeNet). It is not obvious that techniques applicable to convolutional and fully connected layers in CNNs will also apply to element-wise multiplication/addition operations (layers) in LSTMs. It seems more apt to call the proposed techniques memory protection for CNNs only. Additionally, Section 6.1 asserts that these chose CNN models cover a variety of CNNs. Providing more reasons to explain that would justify the choice of CNN models better. d) Section 6.2 illustrates the point in b) above. By reducing 3000 large values (beyond [-64, 63]) in the first seven positions to fit in 6-bits (+1-bit sign) the accuracy of the networks considered remains unchanged. This begs the question, why can’t all the weights be reduced to 6-bits (+1-bit sign) without any drop in accuracy. And if it is possible to reduce the design to fewer bits why use 8-bit implementation at all. In an extreme scenario, a network which has negligible drop in accuracy with 4-bit weights would be forced to use 8-bit precision to use techniques proposed in this paper. Traditional ECC on such a 4-bit network (12.5% penalty on 50% of space) would cost less space than using new proposed zero-space ECC with identical reliability guarantees. e) Lastly, the paper mentions in the abstract that the proposed techniques can be applied to safety-critical systems (e.g. autonomous vehicles). However, arranging weights such that their spatial distribution is fixed (e.g. large weight sitting on last byte in 64bit data block always) will make such systems susceptible to security threats. With that concern in mind the proposed solutions would require some more work before applying to safety-critical systems. Minor Issues a) Neither Figure 3, or the text referencing it in Section 5 clearly explain the definition of fault rates used to create the figure. My guess would be number of bit flips on a target bit in n (e.g. 10000) inferences. b) Fault rate definition should be added for Table 2 as well. c) Few minor typos on following lines – 60, 63, 211 and 247. Apart from above concerns paper has explained the ideas clearly, good job! Author Feedback Comments a) The authors justify well how memory protection for weights is more pressing compared to input/output tensors. At the same time, in-place zero-space ECC requires a specific memory arrangement wherein all 8-bit words have an error-check bit, which is connected to ECC hardware with additional wiring. Implying that an inference engine that wishes to adapt the proposed techniques has three options to manage the input/output tensor dataflow. 1) Use a separate memory (relatively small) with (/preferably without) ECC hardware for input/output tensors. 2) Rely on a processor to control the input/output tensor dataflow (It is possible this is already in place, but then 12.5% savings on the inference engine memory area might be much smaller at the system level.) 3) Somehow manage to store input/output tensors in the same memory as weights and yet read their values correctly despite the weight specific ECC hardware in place. Unless, the authors somehow demonstrate option 3 works, which might have additional cost of its own, the inference engine will have to incur the cost of options 1/2. Considering these costs raises the question if the 12.5% area savings of zero-space technique (most effective) worth it. b) Based on the author feedback I can see that 8-bit quantization offers a sweet spot in terms of ease of implementation (through many industry-supported DNN libraries) and improving latency without compromising much on accuracy. Keeping this in mind it would be befitting to reword Section 3 of the paper to clearly communicate the above reasons for choosing 8-bit quantization. Labeling 8-bit quantization as the de facto step before deployment or most resilient to memory faults seems to ignore the possibility of using even lower precision for certain networks/scenarios. c) Appreciate the authors delineating their choice of CNNs for their experiments, and changing their application scope from DNNs to CNNs. d) As well noted by R3, the techniques proposed in the paper rest on a couple of empirical observations: 1) CNN accuracy does not degrade with quantization to 8bits 2) CNN weights have a distribution wherein a negligible chunk of weights lie in [64, 128) range (absolute value of weights) The above two observations through empirically common, need to be validated before applying these techniques to a new CNN.

Reviewer 3



1. This paper is mainly based on observations and empirical results. For example, the authors should provide more details in the regularization step of regulated QAT process in Section 4.1. If we simply clip the weights, then how is the convergence guaranteed? The authors should elaborate more on theoretical analysis. 2. It seems the paper is off the scope of NeuraIPS. The authors should consider submitting the paper to EDA conferences such as DAC. 3. In Section 5, it is not clear why majority-vote protection is better used under low error rate scenarios. 4. The experiment section is extremely vague. The authors claim that “We use the ImageNet dataset [3] (ILSVRC 2012) for model training and evaluation.” However, the only results based on ImageNet are in Figure 4(b) and 5(b). Are the results in Table 2 based on ImageNet or Cifar10? Additionally, are the models trained from scratch or finetuned? The accuracy curves in Figure 5 look strange to me. For example, the accuracy of SqueezeNet on ImageNet increases from ~10% to ~55% within 2 iterations and the accuracy of VGG16 on Cifar10 is already over 93% at epoch 0. 5. With a thorough look into the codes, I did not find the training module for the three selected models (i.e., VGG16, ResNet18, SqueezeNet) on ImageNet. The authors should provide a description of their codes. Minor issues: 1. In Figure 1, the percentage row, shouldn’t the range be [-128, 128]? 2. There are many typos to be corrected. For example, in the last paragraph of Section 2, “mgeneral” should be “general”.

[Author Response · NeurIPS 2019]

We thank the reviewers for insightful comments. We focus on answering the following concerns from reviewers:

**(R1) Extend the evaluation to even larger models.** Your recognition of our work is much appreciated. We plan to extend our method to more models and include the results as appendix in the final version of the paper.

**(R2) Memory faults against its input/output tensors.** Our work focus on protecting weights for two reasons. Firstly, weights are usually kept in the memory. The longer they are kept, the higher the number of bit flips they will suffer from. This easily results in a high fault rate (e.g. 1e-3) for weights. Input/output tensors, however, are useful only during an inference process. Given the slight chance of having a bit flip during an inference process (usually in milliseconds), protecting input/output tensors is not as pressing as protecting weights. Secondly, previous work [1] have shown that input/output tensors are much less sensitive to faults compared with weights.

**(R2) Protection for lower than 8-bit.** We agree with the reviewer that the optimal bit width for a network depends on its weight distribution and might be lower than 8. However, we have observed that 8-bit quantization is a prevalent, robust, and general choice to reduce model size and latency while preserving accuracy. In our experiments, both activations and weights are quantized to 8-bit. Existing libraries that support quantized DNNs (e.g. NVIDIA TensorRT, Intel MKL-DNN, Google's GEMMLOWP, Facebook's QNNPACK) mainly target for fast operators for 8-bit instead of lower bit width. With that said, we consider extending our protection approaches to lower bit width as future work.

**(R2) DNN Scope and CNN choices.** We agree with the reviewer about changing DNNs to CNNs and will make the change in the paper. We choose these CNN models as representatives because: 1) VGG is a typical CNN with stacked convolutional layers and widely used in transfer learning because of its robustness. 2) ResNets are representatives for CNNs with modular structure (e.g. Residual Module) and are widely used in advanced computer vision tasks such as object detection. 3) SqueezeNet has much fewer parameters and represents CNNs that are designed for mobile applications. We will add the explanations into the paper.

**(R3) Theoretical analysis.** As mentioned in the paper (line 120), the optimization problem WOT tries to solve can also be addressed using an ADMM-based approach similar to [2] that has the theoretical convergence guarantee. However, our empirical explorations have shown that WOT can recover the models' accuracy with less number of iterations. We will provide the comparison between the ADMM-based approach and WOT to supplement the paper.

**(R3) Majority vote for low fault rate settings.** When the fault rate is high (e.g. 1e-3), there is about 1% - 5% accuracy drop even with majority vote protection. As the accuracy drop with majority vote protection in large fault rate settings is significant, it is better used in low error rate scenarios.

**(R3) Experiment settings.** Only results in Figure 3, Figure 4(a) and Figure 5(a) are based on Cifar10. All the other results reported in the paper including Table 2 are collected using ImageNet. For Figure 5, WOT is applied to pre-trained models; the models are finetuned instead of trained from scratch. That's why WOT takes a small number of epochs on Cifar10 or iterations on ImageNet to recover the models' accuracy. Pre-trained SqueezeNet, after clipping the weights to [-64, 63], suffers from a large accuracy drop (56.99% dropped down to around 11.0%). Its accuracy is gradually recovered to 57.01% after fintuning with WOT for 30K iterations on ImageNet. Pre-trained VGG16 is more robust to weight clipping (93.75% dropped down to around 93.4%). Its accuracy is gradually recovered after 16 epochs' training using WOT on Cifar10. We will clarify the settings in the paper.

**Answers to minor issues.**
– (R2) Fault rate/bit error rate is the ratio between the number of bit flips experienced before correction is applied and the total number of bits.
– (R3) In Table 1, the values in percentage rows are absolute values. So the range is [0, 128] instead of [-128, 128].
– (R3) As pre-trained models are used, we don't have a script for training the models from scratch on ImageNet. However, we did include the script for WOT, i.e., regulated training. Because our pre-trained models come from TorchVision, the same script can be reused for the three models by changing the model name in the script.

# References

[1] Brandon Reagen, Udit Gupta, Lillian Pentecost, Paul Whatmough, Sae Kyu Lee, Niamh Mulholland, David Brooks, and Gu-Yeon Wei. Ares: A framework for quantifying the resilience of deep neural networks. In *2018 55th ACM/ESDA/IEEE Design Automation Conference (DAC)*, pages 1–6. IEEE, 2018.

[2] Ao Ren, Tianyun Zhang, Shaokai Ye, Jiayu Li, Wenyao Xu, Xuehai Qian, Xue Lin, and Yanzhi Wang. Admm-nn: An algorithm-hardware co-design framework of dnns using alternating direction methods of multipliers. In *Proceedings of the Twenty-Fourth International Conference on Architectural Support for Programming Languages and Operating Systems*, pages 925–938. ACM, 2019.


[Meta-Review · NeurIPS 2019]

As ML is being used in more places, including mission critical systems, it is important to pay attention to corner cases that could fail. In this paper the authors study memory faults and present a solution for this problem when neural networks are being used. The solutions presented are straight forward however the novelty comes from introducing the problem to the ML communitty and by presenting solutions that take into account the specifics of the ML task. Therefore, this work can ignite an interesting research direction.